# Chemical Solution Deposition of Ordered 2D Arrays of Room-Temperature Ferrimagnetic Cobalt Ferrite Nanodots

**DOI:** 10.3390/polym11101598

**Published:** 2019-09-30

**Authors:** Jin Xu, Justin Varghese, Giuseppe Portale, Alessandro Longo, Jamo Momand, Ali Syari’ati, Jeroen A. Heuver, Petra Rudolf, Bart J. Kooi, Beatriz Noheda, Katja Loos

**Affiliations:** 1Zernike Institute for Advanced Materials, University of Groningen, 9747 AG Groningen, The Netherlands; j.xu@rug.nl (J.X.); justinmv@gmail.com (J.V.); j.momand@rug.nl (J.M.); a.syariati@rug.nl (A.S.); j.a.heuver@gmail.com (J.A.H.); p.rudolf@rug.nl (P.R.); b.j.kooi@rug.nl (B.J.K.); 2Netherlands Organization for Scientific Research (NWO), European Synchrotron Radiation Facility (ESRF), DUBBLE-CRG, 38000 Grenoble, France; alessandro.longo@esrf.fr

**Keywords:** chemical solution deposition, block copolymer, 2D arrays, cobalt ferrite, nanodot, ferrimagnetic

## Abstract

Over the past decades, the development of nano-scale electronic devices and high-density memory storage media has raised the demand for low-cost fabrication methods of two-dimensional (2D) arrays of magnetic nanostructures. Here, we present a chemical solution deposition methodology to produce 2D arrays of cobalt ferrite (CFO) nanodots on Si substrates. Using thin films of four different self-assembled block copolymers as templates, ordered arrays of nanodots with four different characteristic dimensions were fabricated. The dot sizes and their long-range arrangement were studied with scanning electron microscopy (SEM) and grazing incident small-angle X-ray scattering (GISAXS). The structural evolution during UV/ozone treatment and the following thermal annealing was investigated through monitoring the atomic arrangement with X-ray absorption fine structure spectroscopy (EXAFS) and checking the morphology at each preparation step. The preparation method presented here obtains array types that exhibit thicknesses less than 10 nm and blocking temperatures above room temperature (e.g., 312 K for 20 nm diameter dots). Control over the average dot size allows observing an increase of the blocking temperature with increasing dot diameter. The nanodots present promising properties for room temperature data storage, especially if a better control over their size distribution will be achieved in the future.

## 1. Introduction

High-density ordered arrays of magnetic nanodots have attracted much interest due to their potential applications in areas such as nano-scale electronic devices, high-density data storage media and biochips [1,2,3,4,5]. Due to its good chemical and thermal stability, large magnetocrystalline anisotropy and large magnetostriction [6,7,8], the inverse spinel cobalt ferrite (CFO) has been recognized as a promising candidate for such arrays. Although widely fabricated in nanocrystal forms, ordered CFO nanodot arrays supported by substrates were not extensively studied. The reported fabrication methods for CFO nanodot arrays include thin-film patterning [9], nanoparticle self-assembly [2,10,11] and templated deposition [12,13,14]. Thin-film patterning involves a combination of lithography and etching steps, which typically require expensive lithography equipment and/or multiple processing steps. In nanoparticle self-assembly, nano arrays are directly formed from pre-synthesized nanoparticles. This approach faces challenges in mechanical stability and large-area single layer formation. In the templated deposition, the pattern is generated during the material deposition. The previously reported templated deposition approaches for CFO nanodots either require expensive deposition techniques such as pulsed laser deposition (PLD) [12,14], or costly templating techniques such as electron beam lithography [13]. A low-cost fabrication method which does not suffer from the aforementioned drawbacks is highly demanded. Block copolymer thin films have been widely used as templates for nanostructure deposition [15,16] and can be combined with low-cost solution deposition methods. One example is polystyrene-*block*-poly(ethylene oxide) (PS-*b*-PEO) templated chemical solution deposition, which has been used to fabricate a variety of oxide nanodots [17,18,19,20,21,22]. It does not suffer from the aforementioned drawbacks and is low-cost, straight forward and versatile. It exploits the fact that metal ions can selectively form a complex with PEO and hence be localized in the PEO phase [23,24]. Ordered arrays of PEO cylinders either standing up or laying on the substrate can be induced by solvent vapor annealing (SVA) [25,26,27,28,29,30]. Although highly ordered arrays of superparamagnetic Fe_3_O_4_ and Fe_2_O_3_ nanodots [18,20] have been produced, the blocking temperature (*T*_b_), at which the nanodots go through the transition from ferrimagnetic to superparamagnetic, is relatively low (115 K for 20 nm diameter dots), which largely limits their applications in memory storage devices. Raising the blocking temperature to above room temperature is, thus, key to create application opportunities. Here we extend this deposition approach to cobalt ferrite (CFO) nanodots. Various dot sizes and separations were obtained from PS-*b*-PEO thin films with different phase dimensions. Owing to the larger magnetic anisotropy of CFO, T_b_ of the obtained nanodots are greatly enhanced compared to the values for Fe_3_O_4_ [20]. In addition, scanning electron microscopy (SEM) and X-ray absorption fine structure spectroscopy (EXAFS) studies across different preparation steps shed more light on the structural evolution during the preparation.

## 2. Materials and Methods 

All materials were used as received without further purification. Si wafers with (100) surface planes and approximately 2 nm native oxide surface layer were purchased from Microchemicals GmbH (Ulm, Germany) and used as substrates. Iron (III) acetylacetonate (99.9%), cobalt (II) acetylacetonate (97%) and CoFe_2_O_4_ powder (30 nm particle size, 99% trace metals basis) were purchased from Sigma-Aldrich (Zwijndrecht, The Netherlands). All solvents used were of ACS reagent grade. Polystyrene-*block*-poly(ethylene oxide) (PS-*b*-PEO) with various block lengths was purchased from Polymer Source, Inc. (Dorval (Montreal) Quebec, Canada). The different block lengths studied and their naming codes in this article are listed in Table 1.

Si wafers were cut into 1 cm × 1 cm pieces and cleaned by ultrasonication in acetone and then toluene for 15 min each and dried with compressed air. PS-*b*-PEO BCPs were dissolved in toluene and stirred for 12 h, yielding a 1 wt % solution. The solutions were filtered (0.2 mm, PTFE filter) and spin-coated onto the silicon wafers at 2900 rpm for 30 s, with an acceleration rate of 1450 rpm/s. The films were subsequently transferred into a closed 40 mL vessel containing one or two solvent reservoirs (3 mL each). The two solvent reservoirs were used to generate a mixed vapor inside the SVA chamber. The vessel was kept in an oven at 50 °C for 2 h in toluene vapor, or 1 h in the mixed vapor, to enable the self-assembly of the polymer chains. 

Iron (III) acetylacetonate and cobalt (II) acetylacetonate were first dissolved separately in anhydrous methanol and combined afterward to yield an 11 mg/mL precursor solution. After solvent vapor annealing, BCP thin films were immersed in the precursor solution for 40 min at 40 °C and dried with compressed air afterward. The remaining precursor salt on the backside of the wafers was carefully wiped off with ethanol-wetted tissues. The ion-incorporated films were treated with UV/ozone in a Novascan PSDP UV/Ozone cleaner (Boone, IA, USA) to pre-oxidize the precursor and remove the major part of the polymer templates. A two-step high-temperature treatment was subsequently performed in air. The samples were heated up with 10 °C/min rate to 600 °C, kept at temperature for 30 min, then heated to 950 °C with the same rate and kept there for 1 h, before being cooled naturally in air. The fabrication procedure is schematically illustrated in Figure 1.

Atomic force microscopy (AFM) measurements were carried out on a Veeco Dimension V scanning probe microscope (Plainview, NY, USA). Veeco RTESPW tips (resonant frequency 267–294 kHz, spring constant 20–80 N/m) were used in the tapping mode. Software WxSM (Version 4.0 Beta 9.1, Madrid, Spain) was used to analyze the obtained AFM images [31]. SEM images were obtained with a Philips XL30 ESEM, using 5 kV (20 kV) acceleration voltage for BCP films (CFO nanostructures). To reduce the charging effect, the SEM samples were typically coated with a 4 nm Pt/Pd (80:20) layer via Ar^+^ sputtering.

Grazing incidence small-angle X-ray scattering (GISAXS) was performed at the MINA beamline at the University of Groningen. The instrument is based on a Cu rotating anode (X-ray wavelength of λ = 0.15413 nm, photon energy of 8 keV) and has a beam dimension of about 0.5 mm. The used sample-to-detector distance is about 3 m and incident angles ranging from 0.15° to 0.25° were used. Two-dimensional patterns were acquired using a Vantec2000 Bruker detector (Leiderdorp, The Netherlands) and 2D GISAXS patterns are presented as a function of the horizontal *q*_y_ and vertical *q*_z_ scattering vector, with $q_{y}=\frac{2\pi }{\lambda }\left[\sin\left(2\theta_{f}\right)\cos\left(\alpha_{f}\right)\right]$ and $q_{z}=\frac{2\pi }{\lambda }[\sin\left(\alpha_{i}\right)+\;\sin\left(\alpha_{f}\right)]$, where $2\theta_{f}$ and $\alpha_{f}$ are the horizontal and vertical scattering angles, respectively. One-dimensional intensity cuts along the horizontal *q*_y_ and the vertical *q*_z_ directions were extracted from the 2D diffraction pattern using the FITGISAXS program in Igor Pro [32]. The fit of the 1D intensity cuts was achieved using the same program and assuming a hemispherical shape to describe the nanodots supported on the SiO_2_ substrate [32,33]. The fitted variables were the scale factor, dot diameter, dot height, dot separation, the full width half maximum (FWHM) of the Gaussian distribution for the dot diameter, and the positional disorder factor which determines the peak width in the horizontal cuts and is related to the domain size. The dot diameter and the dot height were extracted by fitting the *q*_y_ and *q*_z_ cuts, respectively. In order to achieve the best fitting results, the 1D cuts were first fitted using a monodisperse model and later by introducing a dot size polydispersity. After 1D fitting was achieved, fitting of the 2D patterns was performed. The polydispersity in the dot diameter was taken into account for SEO-49k, SEO-89k and SEO-136k. In contrast, the SEO-26k dots were only fitted using the monodisperse model.

Transmission electron microscopy (TEM) analysis was performed with a JEOL 2010F (Nieuw-Vennep, The Netherlands), operated at 200 kV. TEM specimens were prepared by mechanical grinding and Ar^+^ ion-polishing with a Gatan PIPS II.

Extended X-ray absorption fine structure (EXAFS) spectroscopy measurements were performed at the DUBBLE 26A beamline of the European Synchrotron Radiation Facility (ESRF, Grenoble, France). EXAFS spectra at the Fe K-edge (7112 eV) and Co K-edge (7709 eV) were acquired at liquid nitrogen temperature inside a cryostat, to lower the thermal noise. The standard CFO nanopowder with 30 nm particle size was mixed with cellulose, ground and pressed into a thin pellet, then measured in transmission mode. CFO nanodots grown on Si wafers were measured in the fluorescence mode, at an incident angle of 45°. The absorption coefficient μ(E) was calculated based on the ratio of the transmission or fluorescence intensity to the intensity of the incident X-ray. Using the program Viper, the beginning of the absorption edge *E*_0_ was identified and a smooth background $\mu_{0}\left(E\right)$ was fitted. The fitted background and the jump ∆μ_0_ at the edge were then used to calculate the EXAFS fine-structure function χ (*E*), as
$$\chi \left(E\right)=\frac{\mu \left(E\right)-\mu_{0}\left(E\right)}{∆\mu_{0}}$$

The spectra recorded in energy were converted into χ(k) spectra according to
$$k=\sqrt{\frac{2m\left(E-E_{0}\right)}{\hbar^{2}}}$$
where *k* is the wave vector, *m* is the mass of an electron and ℏ is the reduced Planck constant. The k^2^χ(k) spectra and their Fourier transform in R-space are the forms presented in this work, where R is the distance between the absorbing atom and its relevant neighboring atom.

GNXAS software was used to model the atomic environment around the absorbing atom. In this approach, the local atomic arrangement around the absorbing atom is decomposed into model atomic configurations containing 2, 3, ..., *n* atoms. The theoretical EXAFS signal χ(*k*) is given by the sum of the two-body γ^2^, γ^3^, ..., γ*^n^*, and η^2^, η^3^, ..., η*^n^* three-body contributions, respectively, which take into account all the possible single and multiple scattering (MS) paths between the *n* atoms. The fitting of χ(*k*) to the experimental EXAFS signal allows refining the relevant structural parameters of the different coordination shells. The suitability of the model is also evaluated by comparison of the experimental EXAFS signal Fourier transform (FT) with the FT of the calculated k^2^χ(*k*) function. The fit parameters that were allowed to vary during the fitting procedure were the coordination numbers (CN), the distances R(Å), the Debye–Waller factors (σ) and the angles of the η*^n^* contributions. The threshold energy *E*_0_ was defined at 7.7089 and 7.112 keV for the Co and Fe K-edge, respectively.

According to the atomic arrangement of CFO in the inverse-spinel structure (space group Fd-3 m; a = b = c = 8.400Å) tetrahedral (site 1) and octahedral (site 2) sites are present in the structure and interconnected to each other. Co and Fe atoms reside at the centers of the tetrahedrons and the octahedrons, whereas O atoms occupy the corners. To fit the structure to the spectra, if we take the Co edge as an example, four two-body configurations γ*^2^* corresponding to the four-fold Co–O distance at ~1.9 Å, the six-fold Co–O at 2.06 Å, the six-fold Co–Co at 2.97 Å, and the twelve-fold Co–Co at 3.48 Å were considered. To model the higher shells, three three-body η*^n^* configurations, arising from the three Co–O···M or O–Co···M alignments at 123°, 153° and 120° with Co···M (M = Co or Fe) long distances at ~3.64 Å (four-fold), ~5.14 Å (twelve-fold), and ~5.46 Å (eight-fold), respectively, were taken into account. The model used to analyze the data both at the Fe and Co K-edge, and the paths described are sketched in Appendix A.

X-ray photoemission spectroscopy (XPS) measurements were performed with a Surface Science SSX-100 ESCA instrument with a monochromatic Al Kα X-ray source (*h*_ν_ = 1486.6 eV). The measurement was done at a pressure below 5 × 10^−9^ mbar. The spot size was 1000 μm. The energy resolution was set to 1.26 eV and the electron take-off angle with respect to the surface normal was 37°. The spectra were analyzed using the least-squares curve-fitting program Winspec, developed at the LISE laboratory, University of Namur, Belgium. A Shirley background was used. Binding energies are reported with a precision of ±0.1 eV and referenced to the C1s peak at 284.6 eV [34,35].

Magnetic properties of the samples were investigated with a Quantum Design MPMS XL SQUID magnetometer, using the reciprocating sample option (RSO). The M–T curves were obtained at 100 Oe during heating the sample from 5 to 350 K at a rate of 5 K/min. Prior to the measurements, samples went through either the FC (field-cooling) process, in which samples were cooled from 350 to 5 K (10 K/min) in a 100 Oe field, or the ZFC (zero-field-cooling) process during which samples were cooled from 350 to 5 K (10 K/min) in zero field. The diamagnetic contribution of the substrate was deducted from the obtained M–H loops by fitting and subtracting the linear M–H background with a negative slope. In some cases, the magnetization per cm^3^ is plotted instead of the total magnetization of the sample. This was based on a rough estimation of the total volume (*V*) of the nanodots in one sample using: V = areal density × dot volume. The areal density of dots on the substrate surface was estimated from the SEM images by counting the number of nanodots per 0.5 μm^2^. The dot volume was estimated using the dot height and average dot diameter, assuming a cylindrical dot shape. Thus, the magnetization per unit volume obtained in this way is not an accurate value, but only a rough number for qualitative comparison between samples.

## 3. Results and Discussion

The steps involved in the fabrication of the CFO nanodot arrays are illustrated in Figure 1. Micro-phase separated BCP thin films were obtained by spin coating and annealed in a solvent vapor to increase ordering and form hexagonally-packed PEO cylinders in a PS matrix [18,36]. Fe^3+^ and Co^2+^ ions were selectively loaded into the PEO blocks via immersion in the precursor solution. UV/ozone treatment was then carried out to pre-oxidize the precursor and remove the major part of the polymer templates. A subsequent two-step thermal treatment process at 650 and 900 °C was applied to further oxidize the precursor, remove the rest of the polymer and induce crystallization of the nanodots.

### 3.1. PS-b-PEO Template Preparation

The morphologies of the four block copolymer thin films are shown by the AFM images in the four rows in Figure 2, in which column 1, 2 and 3 depict the film morphology before SVA, after SVA in toluene, and after SVA in a mixed vapor of toluene and water, respectively. The native SiO_2_ layer on the substrate has a stronger interaction with PEO than with PS, due to the hydrogen bonding formation between PEO and the small number of hydroxyl groups on SiO_2_ [37,38]. PS, on the other hand, prefers to wet the free surface, owing to its lower surface tension than the PEO block at 50 °C [20]. However, in the pristine film, a disordered quasi-cylindrical morphology was formed, with PEO cylinders (dark) embedded inside the PS matrix (bright). This observation is in agreement with what was reported in literature and can be explained considering the alignment effect of the field gradient created by fast solvent evaporation during spin-coating [36].

After SVA in toluene for two hours, films from the two shorter BCPs (SEO26k and SEO49k) were found to transform into a highly ordered arrangement of hexagonally-packed cylinders (Figure 2b,e). For the two longer BCPs (SEO89k and SEO136k), however, PS largely replaced PEO on the film surface, yielding a PS-rich surface (Figure 2h,k). Although the solubility parameter differences of PS-water ($\delta_{H_{2}O}-\delta_{PS}=47.9-18.6=29.3{\;\mathrm{MPa}}^{1/2}$) and PEO-water ($\delta_{H_{2}O}-\delta_{PEO}=47.9-20.2=27.7\;{\mathrm{MPa}}^{1/2}$) [39] are both large, PEO can easily dissolve in water due to the hydrogen bonding formation. Adding water into the SVA vapor could, therefore, enhance the free-surface affinity of PEO blocks, forming an arrangement poorer than the hexagonally-packed cylinder arrangement as observed in Figure 2i,l. Based on these results, for CFO nanodot preparation, templates from the two shorter BCPs were treated in toluene vapor for 2 h, while the longer BCPs were treated in toluene/H_2_O mixed vapor for 1 h.

### 3.2. CFO Nanodot Fabrication

Soaking of the BCP thin films in the methanol precursor solution of iron (III) acetylacetonate and cobalt (II) acetylacetonate allows Fe^3+^ and Co^2+^ ions to be selectively loaded into the PEO blocks. A subsequent UV/ozone (UVO) treatment was applied to simultaneously decompose the polymer template and pre-oxide the ions located within the PEO cylindrical domains. The effect of UVO treatment time on the nanodot fabrication is demonstrated by the SEM images of the ion-loaded SEO-89k templates after UVO (see Figure 3). Before UVO treatment (0 min), soaking in the precursor solution did not disrupt the template morphology (Figure 3a). The appearance of protrusions in Figure 3b gives the first indication that the metal ions have undergone a chemical reaction (i.e., became oxidized after 10 min UVO treatment). The residual polymer was further decomposed, and the inorganic dots became clearer and clearer in the SEM images as the treatment time increased to 30 min and further to 60 min. After UVO treatment, nanodots were thermal-annealed (TA) at 600 °C for 30 min, then at 950 °C for 1 h. Without UVO, the ordered structure of the polymeric template was disrupted during TA before the polymer decomposed, resulting in a film of polydisperse CFO nanoparticles with a lower degree of order, as illustrated by Figure 3e.

SEM images of the nanodots prepared from four BCP templates on silicon substrates are shown in Figure 4a. Except for that from the SEO-26k template, nanodot arrays prepared from the other three BCPs all made excellent replications of the PEO cylinder arrays in the original BCP templates. In SEO-26k-templated samples, the dot arrangement appears irregular with many dots missing, while for the SEO-49k-templated samples a much more uniform distribution of dots is observed. The same is true for samples with the other two BCP templates. It seems reasonable to assume that ions at the positions of those missing dots were carried away when the template decomposed, a phenomenon, which seems to occur preferentially for smaller and more close-packed dots.

A precise determination of the diameter of the dots from the SEM images is hampered by the size modification induced by the conductive surface coating. TEM measurements could provide precise dot shape parameters but involve time-consuming and disruptive sample preparation procedures. Moreover, SEM informs only on the local dot arrangement at a selected spot of a few micrometers. GISAXS instead does not require special sample preparation, is undisruptive and relatively time-efficient, and is, therefore, the technique of choice to study the average dot arrangement in a large area (several mm^2^) with simultaneous estimation of the object diameter and height. This estimation is particularly relevant here since the nanodots grow close to each other and AFM tips very often cannot reach the bottom of the space between the dots, making the height parameters from AFM unreliable. 

The experimental and simulated 2D GISAXS patterns are depicted next to the corresponding SEM images in Figure 4b. The corresponding 1D patterns along the *q*_y_ axis are plotted in Figure 4c, together with the best-fit curves (in red). Dots from SEO-49k give three diffraction signals located at 0.20, 0.35 and 0.40 nm^−1^, which correspond to row to row distances of 31.4, 18.0 and 15.7 nm, respectively. The relationship between the three peak positions is 1:1.73:2.00, which is approximately $1:\sqrt{3}:2$, suggesting a hexagonal packing arrangement of the dots in the plane parallel to the substrate. On the contrary, in samples produced from SEO-89k and SEO-136k templates only one broad peak is visible. This is consistent with the poor ordering of the BCP templates from high molecular weight polymers. In the case of the samples templated with SEO-26k, close inspection reveals two distinct lattices, one well-ordered lattice giving rise to the sharp peak located at 0.22 nm^−1^ that originates from the interference between scattering from the neighboring rows of dots, and another broad and weak peak (indicated by the blue arrow) located at around 0.12 nm^−1^, which we propose to associate to scattering from the superlattice formed by groups of dots, separated by vacancies from missing dots. The distance corresponding to this second peak is around 52.9 nm, suggesting an average distance of around 1.8 rows between the groups of dots in the superlattice.

The GISAXS data could be successfully simulated using a model of hemispherical CFO particles supported on Si (see Figure 4b). The excellent agreement between the experimental and simulated curves along the *q*_z_ direction (Appendix A) and the good prediction of the first order peak position along the *q*_y_ direction indicate a reasonable selection of the modeling parameters. Since the model for the SEO-26k-templated dots does not take into account the missing dots, the simulated curve does not include the second peak (indicated by the blue arrow) at low angles. The discrepancy at low angles for the SEO-49k-templated dots most probably originates from the undesired precipitates on the sample surface during sample preparation. The best fit results for the horizontal intensity cuts and for the vertical intensity cuts are reported as the red curves in Figure 4 and Appendix A, respectively. The good quality fit of the GISAXS vertical cuts along the *q*_z_ direction also allows for a precise deduction of the dot height. The structural parameters extracted from fitting the GISAXS patterns are summarized in Table 2 (more details in Appendix A).

### 3.3. Structural Analysis of CFO Nanodots

Figure 5 shows the TEM images of the SEO-49k-templated dots on the Si substrate, prepared using the same conditions as above, except for a soaking time of 2 min. Such a short soaking time yielded dots with smaller height and diameter than the usual 40 min-soaked samples. Each dot is a single crystal, as seen from the high magnification images in the insets. However, crystalline orientations of different nanodots are not unified. This makes the dot arrays practically polycrystalline.

Since the TEM images only analyze a small number of dots and the small dot size and polycrystalline orientation makes X-ray diffraction characterization of the crystalline structure difficult, we chose X-ray absorption fine structure spectroscopy (XAFS) to gain insight into the detailed structure of the nanodots.

The k^2^χ(k) spectra at the Co and Fe K-edge for samples at different preparation stages are presented in Figure 6a and Figure 6c, respectively. Their corresponding FT magnitudes are plotted against R in Figure 6b,d. To make them easier to compare, the plots are shifted vertically, and some are multiplied by a scaling factor, as indicated by the numbers written above the corresponding spectrum. From bottom to top, the spectra are as follows: the ion-loaded BCP template before UVO, after UVO, after UVO and TA at 200 °C, after UVO and TA at 400 °C, after UVO and TA at 600 °C, after UVO and TA at 600 then 950 °C, and the commercial CFO nano powder. The four vertical dashed lines in the Co K-edge FT spectra mark the peak positions for I: Co^1^–O and Co^2^–O, II: Co^2^···M^2^, III: Co^2^···M^1^, Co^1^···M^2^, Co^1^···O and Co^2^···O and IV: Co^2^···M^2^ [40,41,42,43,44], where M represents Co or Fe, and the superscripts 1 and 2 represent respectively the tetrahedral site and the octahedral site (see Appendix A) in the inverse spinel unit cell, respectively. At the Fe K-edge, the peak positions are I: Fe^1^–O and Fe^2^–O, II: Fe^2^···M^2^, III: Fe^2^···M^1^, Fe^1^···M^2^, Fe^1^···O and Fe^2^···O and IV: Fe^2^···M^2^ [40,41,42,43,44].

Due to the lack of mobility below 400 °C, the atoms are not able to rearrange efficiently to form long-range order, only one (Co edge) or two (Fe edge) oscillations are observed. No structure change is evident for the samples annealed at 200 °C. After annealing at 400 °C, two new oscillations around 5 and 6.3 Å^−1^ appear at the Co edge, resulting in the emergence of peak II and III in the FT spectra, indicative of longer-range order. When the temperature is raised to 600 °C, the II to III peak ratio at Co edge increases, more oscillations at both Co and Fe K-edge arise, and peak IV in FT spectra starts to appear, pointing to an improved crystalline structure. After annealing at 950 °C the atomic arrangement is further perfected. The oscillations and FT magnitudes become almost identical to the CFO commercial nanopowder, suggesting an inverse spinel crystalline structure of the nanodots.

The EXAFS spectra from the commercial CFO nanopowder and the final nanodots (annealed at 950 °C) were energy-calibrated, averaged and further analyzed using the GNXAS software. The simulation curves are plotted in Figure 6 as solid red lines, which fit the experimental results very well for both the standard and the nanodots. The two-body and three-body simulation results at the Co K-edge are listed in Table 3 and Table 4, respectively, those at the Fe K-edge are listed in Appendix A. The atomic distances are in good agreement with the literature values for the inverse spinel structure [40,41,42,43,44]. Nevertheless, the coordination numbers (CNs) for both the standard and the nanodots are smaller than the theoretical values for the inverse spinel structure. This is a result of the high surface to volume ratio in nanoparticles, implying that there are more surface atoms with fewer nearest neighbors. The smaller CNs of the nanodots than those in the standard at the Co edge confirm the smaller dimensions of the nanodots (~7 nm × 20 nm) compared to the nanoparticles in the powder used as standard (30 nm × 30 nm). At the Fe edge, the coordination numbers of the nanodots are comparable to the standard. This suggests a larger amount of Co than Fe residing on the dot surface.

XPS measurements were then carried out to analyze the chemical composition and determine the oxidation states. The photoemission spectral lines of the Fe *2p* and Co *2p* core-level regions are plotted in Figure 7a and Figure 7b, respectively. Each of the spin-orbit doublet Fe *2p_3/2_* and Fe *2p_1/2_* peaks is accompanied by a weak shake-up satellite. The peak positions are listed in Table 5. The 7.9 eV splitting between the Fe *2p_3/2_* peak (at a binding energy of 711.4 eV) and its satellite (at 719.3eV) points to a 3+ valence state for Fe; in fact, the typical satellite splitting for Fe^3+^ in oxides is 8~8.8 eV [45,46,47,48,49], while for Fe^2+^ it is around 6 eV [46,50]. The shape and width of the Fe *2p_3/2_* peak indicate the presence of an additional component due to the existence of two nonequivalent lattice sites (i.e., the tetrahedral and octahedral sites in the inverse spinel structure). Deconvolution and fitting of the peaks suggest an approximate occupation of 40% and 60% of the tetrahedral and octahedral sites, respectively.

As expected, similar spectral components were observed at the Co *2p* range: one spin-orbit doublet (Co *2p_3/2_* and Co *2p_1/2_*) followed by relatively strong shake-up satellite peaks. Here the 6.1 eV distance of the satellite (at 786.5 eV) from the Co *2p_3/2_* (at 780.5) testifies to a 2+ valence state, given that the typical satellite splitting for Co^3+^ in oxides is 8.9~9.5 eV [51,52], while for Co^2+^ it is 5~6 eV [46,50]. The deconvolution [53] of the Co^2+^
*2p_3/2_* peak suggests that around 60% of Co^2+^ reside on the octahedral sites while 40% reside on the tetrahedral sites. The ratio between Fe/Co ratio atomic percentages deduced from the photoemission intensities is 1:2. Therefore, the estimated chemical composition of the solid solution is (Co_0.6_Fe_0.7_)(Co_0.9_Fe_1.0_)O_4_, where the parentheses denote the tetrahedral site, and the square brackets denote the octahedral site. Due to the uncertainty of the modeling, this chemical composition is only a rough estimation. However, XPS verified the presence of Co^2+^ and Fe^3+^ on both the octahedral and tetrahedral sites. Giesecke et al. [24] studied complex formation in methanol between monodisperse polyethylene oxide (PEO) and a large set of cations and found that polyvalent cations bind very weakly and give rise to different loadings. The Co/Fe ratio change with respect to the feed could, therefore, be a result of a more efficient complexing of Co^2+^ than Fe^3+^ during template soaking in the precursor. Nevertheless, the present composition is still ferrimagnetic at room temperature, as will be discussed in the next section. Combining the results from XPS and EXAFS, the inverse spinel cobalt ferrite crystalline structure composed of Co^2+^ and Fe^3+^ with an estimated composition of (Co_0.6_Fe_0.7_)[Co_0.9_Fe_1.0_]O_4_ was confirmed. 

### 3.4. Magnetic Properties of the Nanodots

The in-plane magnetization curves from the nanodots with different sizes and separations are shown in Figure 8. At room temperature, all samples exhibit a non-zero remnant magnetization, which makes them potential candidates for room temperature data storage. Data showing the temperature dependence of magnetization are plotted in Figure 9. At low temperature, a large separation between the field-cooling (FC) curves (black) and the zero-field-cooling (ZFC) curves (red) is present, which is additional proof for the presence of net magnetic moment. The two curves get closer as the temperature rises and merge at temperatures close to 350 K. The maxima in the ZFC curves indicate the superparamagnetic transitions at which the nanodots change from ferrimagnetic to superparamagnetic. At this temperature (blocking temperature) the thermal energy is high enough to overcome the magnetocrystalline anisotropy of the nanodots, flipping the dipoles from one easy direction to another so quickly that the instrument cannot detect it. At temperatures higher than blocking temperature the M–H hysteresis disappears, and the remnant magnetization goes to zero. The magnetic parameters extracted from the measurements are listed in Appendix A. The blocking temperatures for all four nanodot sizes are above room temperature, and are much higher than the values (e.g., 150 K for 25 nm nanodots) reported for the magnetite nanodots prepared with a similar procedure [20]. This is a result of the higher magnetocrystalline anisotropy of cobalt ferrite (anisotropy constant K_1_ in the order of 10^6^ ergs/cm^3^) than magnetite (K_1_ in the order of −10^3^ ergs/cm^3^) [54]. As expected, the coercive field (Figure 8) and blocking temperature (Figure 9) are higher for bigger dots, because of the higher magnetocrystalline anisotropy energy (KV) for the larger dot volume (V).

It is worth noting that the measured curves are an overall response from all the dots. At a temperature range lower than the blocking temperature, some dots are already going through the transition to the superparamagnetic state, while some others are not. That is why the transition peak is so broad. This behavior is expected considering the broad size distribution revealed by the GISAXS data. To have more predictable and stable performance as memory storage media it is important to have a sharper transition at a higher temperature. Without having to sacrifice the areal density, this could be achieved by improving the size distribution, increasing the dot thickness, or switching to a material with higher magnetocrystalline anisotropy.

## 4. Conclusions

In conclusion, we report here on the preparation of arrays of ferrimagnetic cobalt ferrite (CFO) nanodots using the block copolymer templated chemical solution deposition approach. Different size and separation of the CFO nanodots were achieved by simply changing the polymer molecular weight. A general characterization flow for such super-thin oxide nanodots, which proved challenging to characterize, was illustrated. The arrangement and the dimensions of the CFO dots were studied by SEM and GISAXS. For some of the produced samples, GISAXS showed successful achievement of ordered 2D hexagonal lattices. Atomic arrangement study by EXAFS on different preparation stages offered rich details on the relationship between UV/ozone treatment, thermal annealing temperatures and events in the dot formation.

The obtained CFO nanodots, with four different sizes, were all ferrimagnetic at room temperature. Due to the higher magnetocrystalline anisotropy [54] the smallest nanodots showed blocking temperature *T*_b_ (310 K) much higher than the value (150 K) of the magnetite nanodots prepared with similar procedures [20]. Blocking temperatures for bigger dots were higher because of the higher magnetocrystalline anisotropy energy (KV) for larger dot volume (V). These relatively high blocking temperatures render them as promising candidates for memory storage applications. Following this direction, one can expect to obtain arrays of ferromagnetic/ferrimagnetic nanodots with even higher blocking temperature by increasing the dot height and using materials with even higher magnetocrystalline anisotropy.

## Figures and Tables

**Figure 1 polymers-11-01598-f001:**
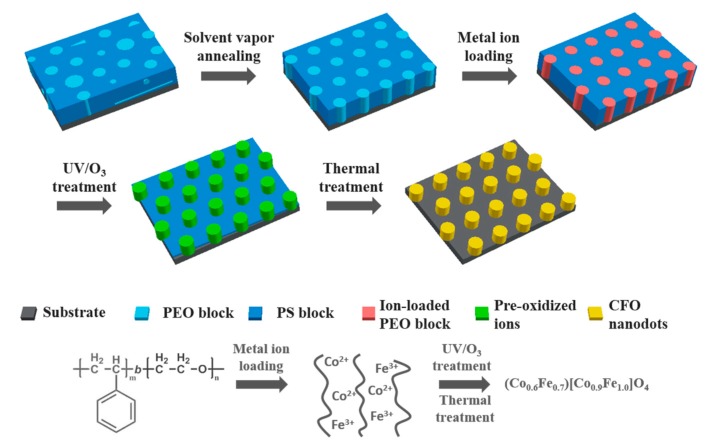
Schematic representation of the fabrication process of the cobalt ferrite (CFO) nanodots. Abbreviations: PEO, poly(ethylene oxide); PS, polystyrene; UV/O, ultraviolet/ozone.

**Figure 2 polymers-11-01598-f002:**
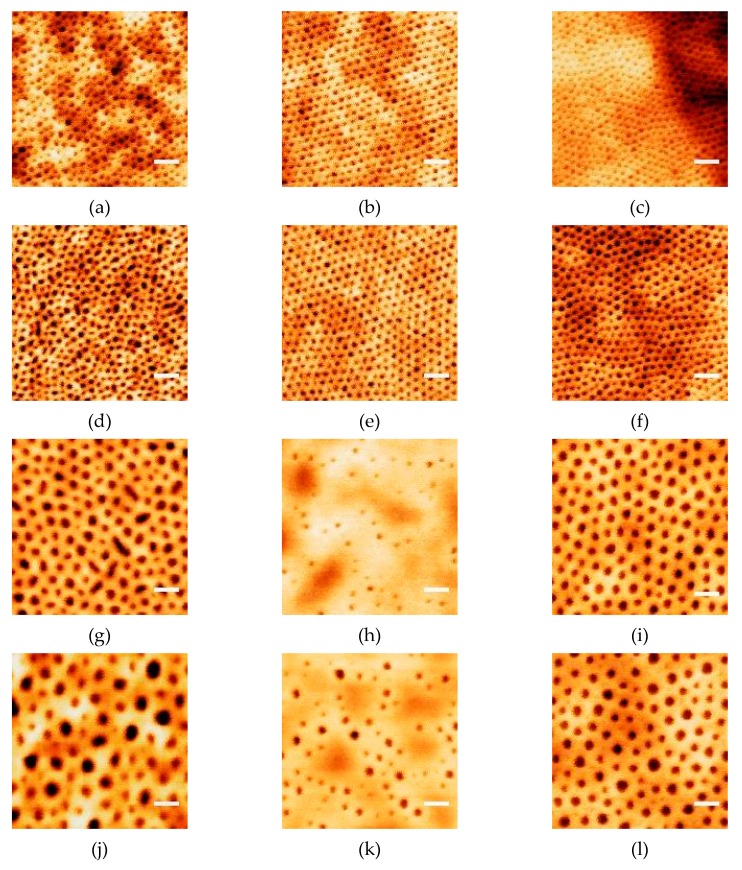
Atomic force microscopy (AFM) images of BCP templates from (**a**–**c**) SEO-26k; (**d**–**f**) SEO-49k; (**g**–**i**) SEO-89k; (**j**–**l**) SEO-136k. Columns 1, 2 and 3 list the film morphology before solvent vapor annealing (SVA), after SVA in toluene at 50 °C for 2 h, and after SVA in a toluene + water mixed vapor at 50 °C for 1 h, respectively. The scale bars are 100 nm.

**Figure 3 polymers-11-01598-f003:**
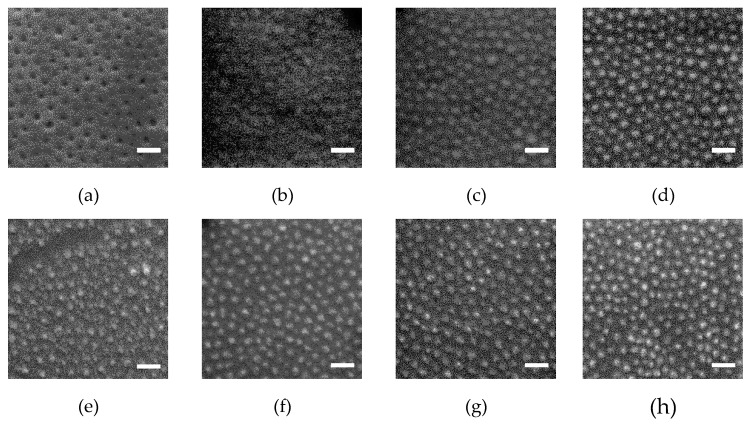
Scanning electron microscopy (SEM) images of the ion-loaded SEO-89k templates after UV/O_3_ treatment. The treatment durations are 0 min for (**a**,**e**); 10 min for (**b**,**f**), 30 min for (**c**,**g**); and 60 min for (**d**,**h**). Images in the first row are from samples before thermal annealing (TA), and in the second row is after TA. The scale bars are 100 nm.

**Figure 4 polymers-11-01598-f004:**
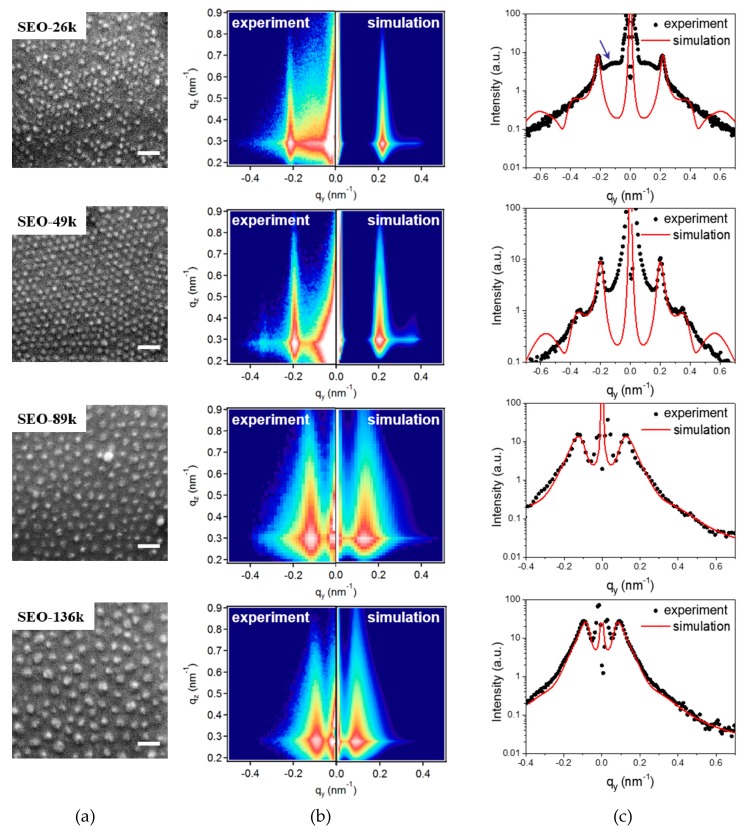
(**a**) SEM images from CFO nanodots (scale bar 100 nm). The text in the white boxes reports the BCP template name (Table 1) used to prepare the CFO dot assemblies. (**b**) The corresponding experimental and simulated 2D grazing incident small-angle X-ray scattering (GISAXS) patterns. (**c**) The corresponding 1D GISAXS patterns along the *q*_y_ direction together with the best-fit curves in red. The blue arrow in the 1D pattern of SEO-26k is a visual aid for a broad diffraction peak.

**Figure 5 polymers-11-01598-f005:**
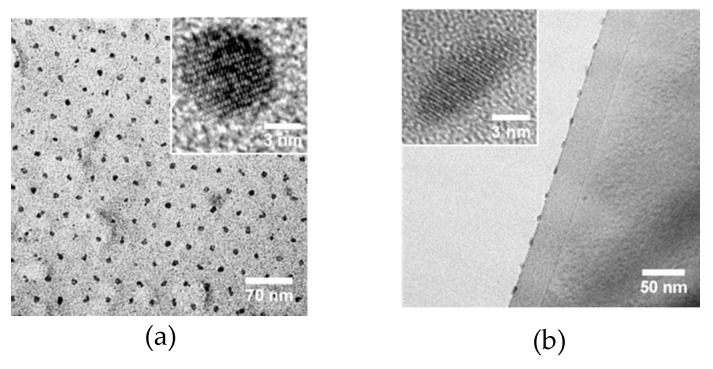
(**a**) Plane view and (**b**) cross-sectional transmission electron microscopy (TEM) images of the CFO nanodots obtained from SEO-49k template. The insets are the high-magnification images.

**Figure 6 polymers-11-01598-f006:**
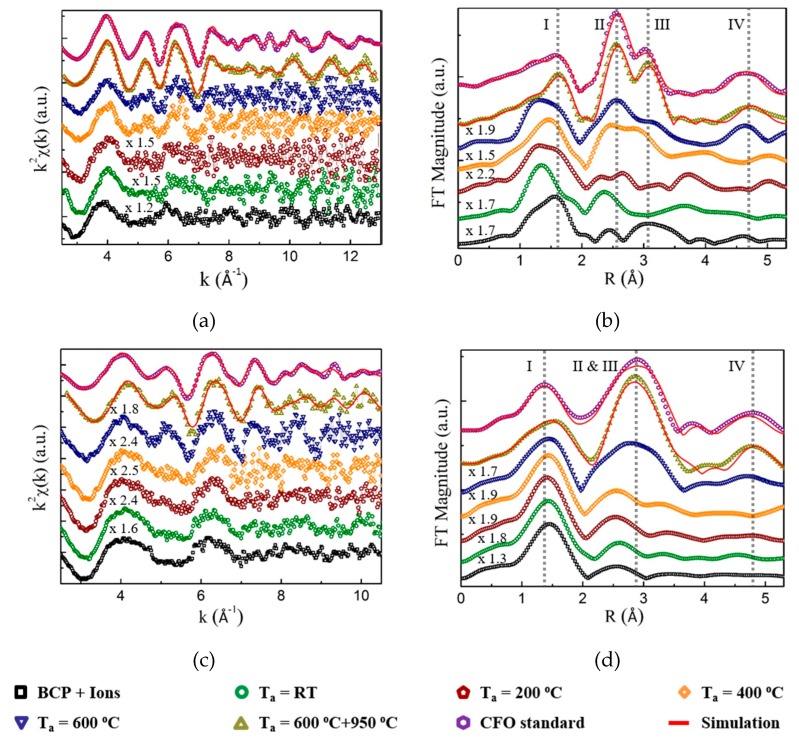
X-ray absorption fine structure spectroscopy (EXAF) signals at (**a**) Co K-edge and (**b**) Fe K-edge. The corresponding Fourier transform magnitudes (FT magnitudes) at (**c**) Co K-edge and (**d**) Fe K-edge for samples at different preparation stages. All plots share the same sample sequence and color code. The solid red lines are the simulation results. The four dashed vertical lines in (**b**,**d**) are used to label the typical peak positions for different shells (see text).

**Figure 7 polymers-11-01598-f007:**
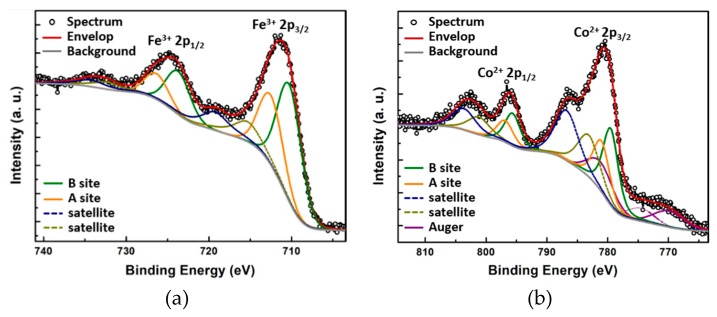
XPS spectra of the (**a**) Fe *2p* and (**b**) Co *2p* core-level regions of CFO nanodots, in which the black open circles indicate the experimental data, the red solid lines are the fitted envelope. The different fitted components are also plotted. A and B sites represent the tetrahedral and octahedral sites in the inverse spinel crystal structure, respectively.

**Figure 8 polymers-11-01598-f008:**
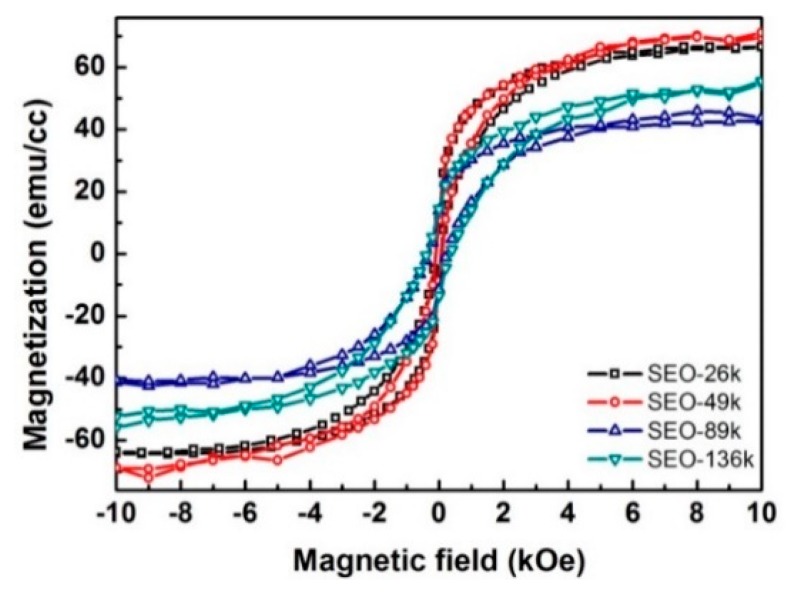
In-plane magnetization curves for nanodots prepared from four different block copolymer templates. All curves were measured at 300 K.

**Figure 9 polymers-11-01598-f009:**
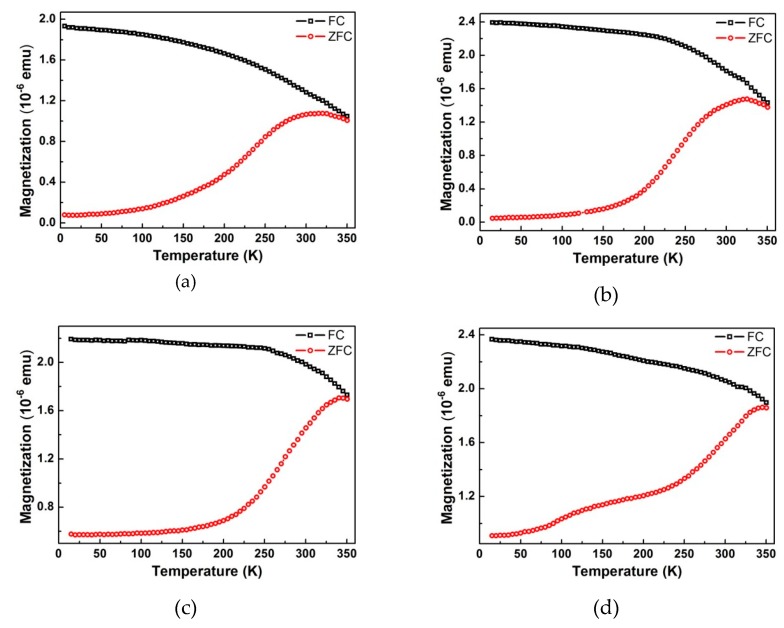
Temperature dependence of the in-plane magnetization for nanodots prepared from (**a**) SEO-26k, (**b**) SEO-49k, (**c**) SEO-89k and (**d**) SEO-136k. The black and red curves are the field-cooling and zero-field-cooling curves, respectively.

**Table 1 polymers-11-01598-t001:** Name codes, number average molecular weights (*M*_n_) and the polydispersity index (PDI) of polystyrene-*block*-poly(ethylene oxide) (PS-*b*-PEO) block copolymers.

Polymer Code	*M*_n,PS_ (kg/mol)	*M*_n,PEO_ (kg/mol)	PDI
SEO-26k	19	6.5	1.09
SEO-49k	38	11	1.06
SEO-89k	63	26	1.07
SEO-136k	102	34	1.18

**Table 2 polymers-11-01598-t002:** Structural parameters deduced from the fit of the experimental GISAXS results, where inter-row distance is the center-to-center distance between two rows of dots.

Template	SEO-26k ^1^	SEO-49k	SEO-89k	SEO-136k
Inter-row distance (nm)	29.0	30.8 ± 4.0	46.5 ± 9.7	53.8 ± 26.9
Diameter (nm)	20.1	20 ± 2.6	26.3 ± 5.5	30 ± 15
Height (nm)	7.3	6.7 ± 0.9	8.6 ± 1.8	9.0 ± 4.5

^1^ Since a monodisperse model was used for fitting, no variations of the simulation results were calculated from the nanodots prepared from the SEO-26k template.

**Table 3 polymers-11-01598-t003:** Two-body simulation results at Co K-edge of the CFO standard and the nanodots annealed at 600 °C and then 950 °C, where CN is the coordination number, *R* is the atomic distance, and σ is the Debye–Waller factor indicating the static and thermal disorder of the shell.

	Nanodots			Commercial Nanopowder		
Shell	CN	*R* (Å)	σ^2^ (Å^2^)	CN	*R* (Å)	σ^2^ (Å^2^)
Co^1^–O	2.5	1.90	0.001	3.6	1.85	0.10
Co^2^–O	4.6	2.05	0.010	5.4	2.04	0.007
Co^2^···M^2^	4.6	2.97	0.005	5.4	2.94	0.005
Co^2^···M^1^	4.6	3.52	0.010	5.4	3.43	0.010
Co^1^···M^2^	7.5	3.44	0001	10.8	3.46	0.003

**Table 4 polymers-11-01598-t004:** Three-body simulation results at Co K-edge of the CFO standard and the nanodots annealed at 600 °C and then 950 °C.

Angles	O–Co^1^–O	O–Co^1^···M^2^	O–Co^2^···M^1^	M^2^ ···Co^2^···M^2^
Nanopowder θ (°)	120	80.93	153.70	120.70
Nanodots θ (°)	113.44	75.50	153.70	120.70

**Table 5 polymers-11-01598-t005:** Binding energy (B.E.), satellite peak position and the satellite splitting (S.S.) energy difference of the XPS spectra. All numbers are in unit eV.

	2*p*_3/2_			2*p*_1/2_		
Range	B.E.	Satellite	S.S.	B.E.	Satellite	S.S.
Fe	711.4	719.3	7.9	724.8	733.4	8.6
Co	780.5	786.5	6.0	796.3	802.6	6.3

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
