# Peer review of "Chemical Solution Deposition of Ordered 2D Arrays of Room-Temperature Ferrimagnetic Cobalt Ferrite Nanodots"

_polymers, 2019, doi:10.3390/polym11101598_

Round 1

Reviewer 1 Report

Journal: POLYMERS

Manuscript ID: polymers-590582

Title: Chemical Solution Deposition of Ordered 2D Arrays of Room-Temperature Ferrimagnetic Cobalt Ferrite Nanodots

This manuscript describes the preparation of arrays of ferrimagnetic cobalt ferrite (CFO) nanodots using the block-copolymer templated chemical solution deposition approach. The size, morphology, atomic arrangement and other properties of the samples were characterized by different characterization techniques in details. In generally, this work is well conducted and organized. The topic is also interesting, with a wide audience. Therefore, I recommend the acceptance after the following minor revisions will be made. I list my detailed comments on this manuscript below.

Comments:

1. There are some format and grammatical errors in this paper. For example, in References section, “37. Grulke, E.A. Solubility … Grulke, E.A., Eds. 1999.”, “MFe2O4”, “CoB2O4”. In Page 3 Line 110, “TEM specimen were prepared by mechanical grinding and Ar+ 110 ion-polishing with a Gatan PIPS II.” should be revised as “TEM specimens were prepared by …”. There are other similar mistakes, which aren’t listed here, please check carefully throughout paper.

2. In Introduction, there is a lack of previous research about 2D arrays of cobalt ferrite (CFO) nanodots in Magnetic properties. The authors should further highlight the novelty of this work.

3. In Results and Discussion section, according to the author's description, the PEO block should be hexagonal, not circular, but the explanatory text below the Figure1 is dark blue. Therefore, the schematic representation should be modified.

4. Figure 8 shows that the coercive field increases as the dot size increases, but the author does not analyse the reason for this trend.

Author Response

There are some format and grammatical errors in this paper. For example, in References section, “37. Grulke, E.A. Solubility … Grulke, E.A., Eds. 1999.”, “MFe2O4”, “CoB2O4”. In Page 3 Line 110, “TEM specimen were prepared by mechanical grinding and Ar+ 110 ion-polishing with a Gatan PIPS II.” should be revised as “TEM specimens were prepared by …”. There are other similar mistakes, which aren’t listed here, please check carefully throughout paper.

Answer: Thank you for your comment. Formatting and grammar were rechecked and corrected.

In Introduction, there is a lack of previous research about 2D arrays of cobalt ferrite (CFO) nanodots in Magnetic properties. The authors should further highlight the novelty of this work.

Answer: Thanks for your advice. Not too much research has been done on 2D arrays of CFO nanodots. The existing reports have heavily focused on the fabrication, with only most common magnetic characterizations included, as in the current work. But to explain the advantage in the magnetic properties of CFO, a few references were added on high magnetostriction and anisotropy (line 36) of CFO thin films or nanofibers, and the introduction was modified to highlight the novelty of this work.

In Results and Discussion section, according to the author's description, the PEO block should be hexagonal, not circular, but the explanatory text below the Figure1 is dark blue. Therefore, the schematic representation should be modified.

Answer: Thanks for your comment. The wrong illustration was corrected in Figure 1.

Figure 8 shows that the coercive field increases as the dot size increases, but the author does not analyse the reason for this trend.

Answer: The reason was analyzed together with Figure 9 in line 406 ~ 408. The coercive field (Figure 8) and blocking temperature (Figure 9) are higher for bigger dots, because of the higher magnetocrystalline anisotropy energy (KV) for the larger dot volume (V).

Reviewer 2 Report

This work reported for a chemical solution deposition method to produce 2D arrays of cobalt ferrite (CFO) nanodots on Si substrates using various experiments such as GI-SAXS, EXAFS, TEM, AFM, SEM, XPS and SQUID, where the experimental results sound good. This work is very interesting and can provide a good method to fabricate 2-dimensional self-assembly of nanodot. Therefore, it was recommended for publication with a minor revision.

In Figure 1, there is a blue line in the legend position. It should be removed. In Figure 2, the numbering (a-l) position is rather strange that should be revised. In Figure 8, the author presents the magnetization curves for nanodots prepared from four different block copolymer. However, it is rather difficult to be recognized. Each curves should be presented separately for visual clarity.

Author Response

In Figure 1, there is a blue line in the legend position. It should be removed.

Answer: Thanks for your feedback. However, this blue line cannot be observed in in our word or pdf version of the manuscript. It is most probably a file formatting issue.

In Figure 2, the numbering (a-l) position is rather strange that should be revised.

Answer: The numbering positions are normal in the word format. Position disorder occurred during pdf conversion.

In Figure 8, the author presents the magnetization curves for nanodots prepared from four different block copolymer. However, it is rather difficult to be recognized. Each curves should be presented separately for visual clarity.

Answer: Thanks for your suggestion. We however consider it important that the curves can be directly compared in one graph and therefore did not change the figure.

Reviewer 3 Report

This manuscript is well written and deals with an interesting manuscript of a highly actual topic in polymer science. The authors employ four different PS-b-PEO diblock copolymers in thin films as templates for the formation of ordered cobalt ferrite nanodots. So, in summary I can highly recommend this manuscript to be published in polymers. In my opinion it would need only some minor revisions prior to publication:

- In the introduction the authors should explain the term ‘blocking temperature’. This is a main result of the authors and it is not so familiar to polymer scientist (even if the authors introduce this further below).

- Maybe the authors provide the information on the different block lengths of the PS-b-PEO copolymers in the main text of the experimental part. This is a basic information and does not necessarily belong to the supporting information.

- page 4 line 160. Put a blank between the number and the unit, e.g. 5 K.

- Figure 2 has some problems with the positions of (d), (e) etc. Maybe this happened during the conversion to the pdf-file.

- Is there a reason why the sample SEO-26k is fitted with the monodisperse model in contrast to the other three samples?

- The authors might provide a reference to T.P. Russell et al. They published the first paper to use block copolymer templates for the formation of ordered inorganic nanostructure arrays in thin films (Science 2000).

Author Response

- In the introduction the authors should explain the term ‘blocking temperature’. This is a main result of the authors and it is not so familiar to polymer scientist (even if the authors introduce this further below).

Answer: Thank you for your valuable suggestion. One-sentence explanation of ‘blocking temperature’ is added to the introduction (line 57).

- Maybe the authors provide the information on the different block lengths of the PS-b-PEO copolymers in the main text of the experimental part. This is a basic information and does not necessarily belong to the supporting information.

Answer: Thank you for your comment. We agree and therefore Table S1 was moved from supporting information to the main text, becoming Table 1.

- page 4 line 160. Put a blank between the number and the unit, e.g. 5 K.

Answer: Formatting was corrected.

- Figure 2 has some problems with the positions of (d), (e) etc. Maybe this happened during the conversion to the pdf-file.

Answer: The numbering positions are normal in the word format. Position disorder occurred during pdf conversion. We will advise the copy editor to watch out for this problem in the galley proof.

- Is there a reason why the sample SEO-26k is fitted with the monodisperse model in contrast to the other three samples?

Answer: Thanks for your comment. Due to the large disorder in the SEO-26k matrix (from the many missing dots), the polydisperse model resulted into a worse fitting compared to the monodisperse model. Therefore, the monodisperse model was used instead of polydisperse model.

- The authors might provide a reference to T.P. Russell et al. They published the first paper to use block copolymer templates for the formation of ordered inorganic nanostructure arrays in thin films (Science 2000).

Answer: Thanks for your advice. The recommended reference has been added (ref. 15, line 49).